# Towards Deployment-Efficient Reinforcement Learning: Lower Bound and Optimality

**Jiawei Huang**\*†**, Jinglin Chen**†**, Li Zhao**‡**, Tao Qin**‡**, Nan Jiang**†**, Tie-Yan Liu**‡

† Department of Computer Science, University of Illinois at Urbana-Champaign
`{jiaweih, jinglinc, nanjiang}@illinois.edu`
‡ Microsoft Research Asia
`{lizo, taoqin, tyliu}@microsoft.com`

## Abstract

Deployment efficiency is an important criterion for many real-world applications of reinforcement learning (RL). Despite the community's increasing interest, there lacks a formal theoretical formulation for the problem. In this paper, we propose such a formulation for deployment-efficient RL (DE-RL) from an "optimization with constraints" perspective: we are interested in exploring an MDP and obtaining a near-optimal policy within minimal *deployment complexity*, whereas in each deployment the policy can sample a large batch of data. Using finite-horizon linear MDPs as a concrete structural model, we reveal the fundamental limit in achieving deployment efficiency by establishing information-theoretic lower bounds, and provide algorithms that achieve the optimal deployment efficiency. Moreover, our formulation for DE-RL is flexible and can serve as a building block for other practically relevant settings; we give "Safe DE-RL" and "Sample-Efficient DE-RL" as two examples, which may be worth future investigation.

## 1 Introduction

In many real-world applications, deploying a new policy to replace the previous one is costly, while generating a large batch of samples with an already deployed policy can be relatively fast and cheap. For example, in recommendation systems (Afsar et al., 2021), education software (Bennane et al., 2013), and healthcare (Yu et al., 2019), the new recommendation, teaching, or medical treatment strategy must pass several internal tests to ensure safety and practicality before being deployed, which can be time-consuming. On the other hand, the algorithm may be able to collect a large amount of samples in a short period of time if the system serves a large population of users. Besides, in robotics applications (Kober et al., 2013), deploying a new policy usually involves operations on the hardware level, which requires non-negligible physical labor and long waiting periods, while sampling trajectories is relatively less laborious. However, deployment efficiency was neglected in most of existing RL literatures. Even for those few works considering this important criterion (Bai et al., 2020; Gao et al., 2021; Matsushima et al., 2021), either their settings or methods have limitations in the scenarios described above, or a formal mathematical formulation is missing. We defer a detailed discussion of these related works to Section 1.1.

In order to close the gap between existing RL settings and real-world applications requiring high deployment efficiency, our first contribution is to provide a formal definition and tractable objective for Deployment-Efficient Reinforcement Learning (DE-RL) via an "optimization with constraints" perspective. Roughly speaking, we are interested in minimizing the number of deployments $K$ under two constraints: (a) after deploying $K$ times, the algorithm can return a near-optimal policy, and (b) the number of *trajectories* collected in each deployment, denoted as $N$, is at the same level across $K$ deployments, and it can be large but should still be polynomial in standard parameters. Similar to the notion of sample complexity in online RL, we will refer to $K$ as *deployment complexity*.

---

\*Work done during the internship at Microsoft Research Asia.

To provide a more quantitative understanding, we instantiate our DE-RL framework in finite-horizon linear MDPs[1] (Jin et al., 2019) and develop the essential theory. The main questions we address are:

> *Q1: What is the optimum of the deployment efficiency in our DE-RL setting?*
> *Q2: Can we achieve the optimal deployment efficiency in our DE-RL setting?*

When answering these questions, we separately study algorithms with or without being constrained to deploy deterministic policies each time. While deploying more general forms of policies can be practical (e.g., randomized experiments on a population of users can be viewed as deploying a mixture of deterministic policies), most previous theoretical works in related settings exclusively focused on upper and lower bounds for algorithms using deterministic policies (Jin et al., 2019; Wang et al., 2020b; Gao et al., 2021). As we will show, the origin of the difficulty in optimizing deployment efficiency and the principle in algorithm design to achieve optimal deployment efficiency are generally different in these two settings, and therefore, we believe both of them are of independent interests.

As our second contribution, in Section 3, we answer Q1 by providing information-theoretic lower bounds for the required number of deployments under the constraints of (a) and (b) in Def 2.1. We establish $\Omega(dH)$ and $\widetilde{\Omega}(H)$ lower bounds for algorithms with and without the constraints of deploying deterministic policies, respectively. Contrary to the impression given by previous empirical works (Matsushima et al., 2021), even if we can deploy unrestricted policies, the minimal number of deployments cannot be reduced to a constant without additional assumptions, which sheds light on the fundamental limitation in achieving deployment efficiency. Besides, in the line of work on "horizon-free RL" (e.g., Wang et al., 2020a), it is shown that RL problem is not significantly harder than bandits (i.e., when $H = 1$) when we consider sample complexity. In contrast, the $H$ dependence in our lower bound reveals some fundamental hardness that is specific to long-horizon RL, particularly in the deployment-efficient setting. [2] Such hardness results were originally conjectured by Jiang & Agarwal (2018), but no hardness has been shown in sample-complexity settings.

After identifying the limitation of deployment efficiency, as our third contribution, we address Q2 by proposing novel algorithms whose deployment efficiency match the lower bounds. In Section 4.1, we propose an algorithm deploying deterministic policies, which is based on Least-Square Value Iteration with reward bonus (Jin et al., 2019) and a layer-by-layer exploration strategy, and can return an $\varepsilon$-optimal policy within $O(dH)$ deployments. As part of its analysis, we prove Lemma 4.2 as a technical contribution, which can be regarded as a batched finite-sample version of the well-known "Elliptical Potential Lemma"(Carpentier et al., 2020) and may be of independent interest. Moreover, our analysis based on Lemma 4.2 can be applied to the reward-free setting (Jin et al., 2020; Wang et al., 2020b) and achieve the same optimal deployment efficiency. In Section 4.2, we focus on algorithms which can deploy arbitrary policies. They are much more challenging because it requires us to find a provably exploratory stochastic policy without interacting with the environment. To our knowledge, Agarwal et al. (2020b) is the only work tackling a similar problem, but their algorithm is model-based which relies on a strong assumption about the realizability of the true dynamics and a sampling oracle that allows the agent to sample data from the model, and how to solve the problem in linear MDPs without a model class is still an open problem. To overcome this challenge, we propose a model-free layer-by-layer exploration algorithm based on a novel covariance matrix estimation technique, and prove that it requires $\Theta(H)$ deployments to return an $\varepsilon$-optimal policy, which only differs from the lower bound $\widetilde{\Omega}(H)$ by a logarithmic factor. Although the per-deployment sample complexity of our algorithm has dependence on a "reachability coefficient" (see Def. 4.3), similar quantities also appear in related works (Zanette et al., 2020; Agarwal et al., 2020b; Modi et al., 2021) and we conjecture that it is unavoidable and leave the investigation to future work.

Finally, thanks to the flexibility of our "optimization with constraints" perspective, our DE-RL setting can serve as a building block for more advanced and practically relevant settings where optimizing the number of deployments is an important consideration. In Appendix F, we propose two potentially interesting settings: "Safe DE-RL" and "Sample-Efficient DE-RL", by introducing constraints regarding safety and sample efficiency, respectively.

---

[1]Although we focus on linear MDPs, the core idea can be extended to more general settings such as RL with general function approximation (Kong et al., 2021).

[2]Although (Wang et al., 2020a) considered stationary MDP, as shown in our Corollary 3.3, the lower bounds of deployment complexity is still related to $H$.

## 1.1 CLOSELY RELATED WORKS

We defer the detailed discussion of previous literatures about pure online RL and pure offline RL to Appendix A, and mainly focus on those literatures which considered deployment efficiency and more related to us in this section.

To our knowledge, the term "deployment efficiency" was first coined by Matsushima et al. (2021), but they did not provide a concrete mathematical formulation that is amendable to theoretical investigation. In existing theoretical works, low switching cost is a concept closely related to deployment efficiency, and has been studied in both bandit (Esfandiari et al., 2020; Han et al., 2020; Gu et al., 2021; Ruan et al., 2021) and RL settings (Bai et al., 2020; Gao et al., 2021; Kong et al., 2021). Another related concept is concurrent RL, as proposed by Guo & Brunskill (2015). We highlight the difference with them in two-folds from problem setting and techniques.

As for the problem setting, existing literature on low switching cost mainly focuses on sub-linear regret guarantees, which does not directly implies a near-optimal policy after a number of policy deployments[3]. Besides, low switching-cost RL algorithms (Bai et al., 2020; Gao et al., 2021; Kong et al., 2021) rely on adaptive switching strategies (i.e., the interval between policy switching is not fixed), which can be difficult to implement in practical scenarios. For example, in recommendation or education systems, once deployed, a policy usually needs to interact with the population of users for a fair amount of time and generate a lot of data. Moreover, since policy preparation is time-consuming (which is what motivates our work to begin with), it is practically difficult if not impossible to change the policy immediately once collecting enough data for policy update, and it will be a significant overhead compared to a short policy switch interval. Therefore, in applications we target at, it is more reasonable to assume that the sample size in each deployment (i.e., between policy switching) has the same order of magnitude and is large enough so that the overhead of policy preparation can be ignored.

More importantly, on the technical side, previous theoretical works on low switching cost mostly use deterministic policies in each deployment, which is easier to analyze. This issue also applies to the work of Guo & Brunskill (2015) on concurrent PAC RL. However, if the agent can deploy stochastic (and possibly non-Markov) policies (e.g., a mixture of deterministic policies), then intuitively— and as reflected in our lower bounds—exploration can be done much more deployment-efficiently, and we provide a stochastic policy algorithm that achieves an $\widetilde{O}(H)$ deployment complexity and overcomes the $\Omega(dH)$ lower bounds for deterministic policy algorithms (Gao et al., 2021).

## 2 PRELIMINARIES

**Notation** Throughout our paper, for $n \in \mathbb{Z}^+$, we will denote $[n] = \{1, 2, ..., n\}$. $\lceil \cdot \rceil$ denotes the ceiling function. Unless otherwise specified, for vector $x \in \mathbb{R}^d$ and matrix $X \in \mathbb{R}^{d \times d}$, $\|x\|$ denotes the vector $l_2$-norm of $x$ and $\|X\|$ denotes the largest singular value of $X$. We will use standard big-oh notations $O(\cdot), \Omega(\cdot), \Theta(\cdot)$, and notations such as $\widetilde{O}(\cdot)$ to suppress logarithmic factors.

## 2.1 EPISODIC REINFORCEMENT LEARNING

We consider an episodic Markov Decision Process denoted by $M(\mathcal{S}, \mathcal{A}, H, P, r)$, where $\mathcal{S}$ is the state space, $\mathcal{A}$ is the finite action space, $H$ is the horizon length, and $P = \{P_h\}_{h=1}^{H}$ and $r = \{r_h\}_{h=1}^{H}$ denote the transition and the reward functions. At the beginning of each episode, the environment will sample an initial state $s_1$ from the initial state distribution $d_1$. Then, for each time step $h \in [H]$, the agent selects an action $a_h \in \mathcal{A}$, interacts with the environment, receives a reward $r_h(s_h, a_h)$, and transitions to the next state $s_{h+1}$. The episode will terminate once $s_{H+1}$ is reached.

A (Markov) policy $\pi_h(\cdot)$ at step $h$ is a function mapping from $\mathcal{S} \to \Delta(\mathcal{A})$, where $\Delta(\mathcal{A})$ denotes the probability simplex over the action space. With a slight abuse of notation, when $\pi_h(\cdot)$ is a deterministic policy, we will assume $\pi_h(\cdot) : \mathcal{S} \to \mathcal{A}$. A full (Markov) policy $\pi = \{\pi_1, \pi_2, ..., \pi_H\}$ specifies such a mapping for each time step. We use $V_h^\pi(s)$ and $Q_h^\pi(s, a)$ to denote the value function

---

[3]Although the conversion from sub-linear regret to polynominal sample complexity is possible ("online-to-batch"), we show in Appendix A that to achieve accuracy $\varepsilon$ after conversion, the number of deployments of previous low-switching cost algorithms has dependence on $\varepsilon$, whereas our guarantee does not.

and Q-function at step $h \in [H]$, which are defined as:

$$V_h^\pi(s) = \mathbb{E}[\sum_{h'=h}^{H} r_{h'}(s_{h'}, a_{h'})|s_h = s, \pi], \quad Q_h^\pi(s, a) = \mathbb{E}[\sum_{h'=h}^{H} r_{h'}(s_{h'}, a_{h'})|s_h = s, a_h = a, \pi]$$

We also use $V_h^*(\cdot)$ and $Q_h^*(\cdot, \cdot)$ to denote the optimal value functions and use $\pi^*$ to denote the optimal policy that maximizes the expected return $J(\pi) := \mathbb{E}[\sum_{h=1}^{H} r(s_h, a_h)|\pi]$. In some occasions, we use $V_h^\pi(s; r)$ and $Q_h^\pi(s, a; r)$ to denote the value functions with respect to $r$ as the reward function for disambiguation purposes. The optimal value functions and the optimal policy will be denoted by $V^*(s; r), Q^*(s, a; r), \pi_r^*$, respectively.

**Non-Markov Policies**   While we focus on Markov policies in the above definition, some of our results apply to or require more general forms of policies. For example, our lower bounds apply to non-Markov policies that can depend on the history (e.g., $\mathcal{S}_1 \times \mathcal{A}_1 \times \mathbb{R}... \times \mathcal{S}_{h-1} \times \mathcal{A}_{h-1} \times \mathbb{R} \times \mathcal{S}_h \to \mathcal{A}$ for deterministic policies); our algorithm for arbitrary policies deploys a mixture of deterministic Markov policies, which corresponds to choosing a deterministic policy from a given set at the initial state, and following that policy for the entire trajectory. This can be viewed as a non-Markov stochastic policy.

## 2.2   LINEAR MDP SETTING

We mainly focus on the linear MDP (Jin et al., 2019) satisfying the following assumptions:

**Assumption A** (Linear MDP Assumptions)**.**   An MDP $\mathcal{M} = (\mathcal{S}, \mathcal{A}, H, P, r)$ is said to be a linear MDP with a feature map $\phi : \mathcal{S} \times \mathcal{A} \to \mathbb{R}^d$ if the following hold for any $h \in [H]$:

- There are $d$ unknown signed measures $\mu_h = (\mu_h^{(1)}, \mu_h^{(2)}, ..., \mu_h^{(d)})$ over $\mathcal{S}$ such that for any $(s, a, s') \in \mathcal{S} \times \mathcal{A} \times \mathcal{S}$, $P_h(s'|s, a) = \langle \mu_h(s'), \phi(s, a) \rangle$.

- There exists an unknown vector $\theta_h \in \mathbb{R}^d$ such that for any $(s, a) \in \mathcal{S} \times \mathcal{A}$, $r_h(s, a) = \langle \phi(s, a), \theta_h \rangle$.

Similar to Jin et al. (2019) and Wang et al. (2020b), without loss of generality, we assume for all $(s, a) \in \mathcal{S} \times \mathcal{A}$ and $h \in [H]$, $\|\phi(s, a)\| \leqslant 1$, $\|\mu_h\| \leqslant \sqrt{d}$, and $\|\theta_h\| \leqslant \sqrt{d}$. In Section 3 we will refer to linear MDPs with stationary dynamics, which is a special case when $\mu_1 = \mu_2 = \dots \mu_H$ and $\theta_1 = \theta_2 = \dots = \theta_H$.

## 2.3   A CONCRETE DEFINITION OF DE-RL

In the following, we introduce our formulation for DE-RL in linear MDPs. For discussions of comparison to existing works, please refer to Section 1.1.

**Definition 2.1** (Deployment Complexity in Linear MDPs)**.**   We say that an algorithm has a deployment complexity $K$ in linear MDPs if the following holds: given an arbitrary linear MDP under Assumption A, for arbitrary $\varepsilon$ and $0 < \delta < 1$, the algorithm will return a policy $\pi_K$ after $K$ deployments and collecting at most $N$ trajectories in each deployment, under the following constraints:

(a)  With probability $1 - \delta$, $\pi_K$ is $\varepsilon$-optimal, i.e. $J(\pi_K) \geqslant \max_\pi J(\pi) - \varepsilon$.

(b)  The sample size $N$ is polynominal, i.e. $N = \text{poly}(d, H, \frac{1}{\varepsilon}, \log\frac{1}{\delta})$. Moreover, $N$ should be fixed a priori and cannot change adaptively from deployment to deployment.

Under this definition, the goal of Deployment-Efficient RL is to design algorithms with provable guarantees of low deployment complexity.

**Polynomial Size of $N$**   We emphasize that the restriction of polynomially large $N$ is crucial to our formulation, and not including it can result in degenerate solutions. For example, if $N$ is allowed to be exponentially large, we can finish exploration in 1 deployment in the arbitrary policy setting, by deploying a mixture of exponentially many policies that form an $\varepsilon$-net of the policy space. Alternatively, we can sample actions uniformly, and use importance sampling (Precup, 2000) to evaluate all of them in an off-policy manner. None of these solutions are practically feasible and are excluded by our restriction on $N$.

# 3 LOWER BOUND FOR DEPLOYMENT COMPLEXITY IN RL

In this section, we provide information-theoretic lower bounds of the deployment complexity in our DE-RL setting. We defer the lower bound construction and the proofs to Appendix B. As mentioned in Section 2, we consider non-Markov policies when we refer to deterministic and stochastic policies in this section, which strengthens our lower bounds as they apply to very general forms of policies.

We first study the algorithms which can only deploy deterministic policy at each deployment.

**Theorem 3.1.** *[Lower bound for deterministic policies, informal] For any $d \geqslant 4$, $H$ and any algorithm $\psi$ that can only deploy a deterministic policy at each deployment, there exists a linear MDP $M$ satisfying Assumption A, such that the deployment complexity of $\psi$ in $M$ is $K = \Omega(dH)$.*

The basic idea of our construction and the proof is that, intuitively, a linear MDP with dimension $d$ and horizon length $H$ has $\Omega(dH)$ "independent directions", while deterministic policies have limited exploration capacity and only reach $\Theta(1)$ direction in each deployment, which result in $\Omega(dH)$ deployments in the worst case.

In the next theorem, we will show that, even if the algorithm can use arbitrary exploration strategy (e.g. maximizing entropy, adding reward bonus), without additional assumptions, the number of deployments $K$ still has to depend on $H$ and may not be reduced to a constant when $H$ is large.

**Theorem 3.2.** *[Lower bound for arbitrary policies, informal] For any $d \geqslant 4$, $H$, $N$ and any algorithm $\psi$ which can deploy arbitrary policies, there exists a linear MDP $M$ satisfying Assumption A, such that the deployment complexity of $\psi$ in $M$ is $K = \Omega(H/\lceil \log_d(NH) \rceil) = \widetilde{\Omega}(H)$.*

The origin of the difficulty can be illustrated by a recursive dilemma: in the worst case, if the agent does not have enough information at layer $h$, then it cannot identify a good policy to explore till layer $h + \Omega(\log_d(NH))$ in 1 deployment, and so on and so forth. Given that we enforce $N$ to be polynomial, the agent can only push the "information boundary" forward by $\Omega(\log_d(NH)) = \widetilde{\Omega}(1)$ layers per deployment. In many real-world applications, such difficulty can indeed exist. For example, in healthcare, the entire treatment is often divided into multiple stages. If the treatment in stage $h$ is not effective, the patient may refuse to continue. This can result in insufficient samples for identifying a policy that performs well in stage $h + 1$.

**Stationary vs. non-stationary dynamics** Since we consider non-stationary dynamics in Assump. A, one may suspect that the $H$-dependence in the lower bound is mainly due to such non-stationarity. We show that this is not quite the case, and the $H$-dependence still exists for stationary dynamics. In fact, our lower bound for non-stationary dynamics directly imply one for stationary dynamics: given a finite horizon non-stationary MDP $\widetilde{M} = (\widetilde{\mathcal{S}}, \mathcal{A}, H, \widetilde{P}, \widetilde{r})$, we can construct a stationary MDP $M = (\mathcal{S}, \mathcal{A}, H, P, r)$ by expanding the state space to $\mathcal{S} = \widetilde{\mathcal{S}} \times [H]$ so that the new transition function $P$ and reward function $r$ are stationary across time steps. As a result, given arbitrary $d \geqslant 4$ and $H \geqslant 2$, we can construct a hard non-stationary MDP instance $\widetilde{M}$ with dimension $\widetilde{d} = \max\{4, d/H\}$ and horizon $\widetilde{h} = d/\widetilde{d} = \min\{H, d/4\}$, and convert it to a stationary MDP $M$ with dimension $d$ and horizon $h = \widetilde{h} = \min\{H, d/4\} \leqslant H$. If there exists an algorithm which can solve $M$ in $K$ deployments, then it can be used to solve $\widetilde{M}$ in no more than $K$ deployments. Therefore, the lower bounds for stationary MDPs can be extended from Theorems 3.1 and 3.2, as shown in the following corollary:

**Corollary 3.3** (Extension to Stationary MDPs). *For stationary linear MDP with $d \geqslant 4$ and $H \geqslant 2$, suppose $N = \text{poly}(d, H, \frac{1}{\varepsilon}, \log \frac{1}{\delta})$, the lower bound of deployment complexity would be $\Omega(d)$ for deterministic policy algorithms, and $\Omega(\frac{\min\{d/4, H\}}{\lceil \log_{\max\{d/H, 4\}} NH \rceil}) = \widetilde{\Omega}(\min\{d, H\})$ for algorithms which can deploy arbitrary policies.*

As we can see, the dependence on dimension and horizon will not be eliminated even if we make a stronger assumption that the MDP is stationary. The intuition is that, although the transition function is stationary, some states may not be reachable from the initial state distribution within a small number of times, so the stationary MDP can effectively have a "layered" structure. For example, in Atari games (Bellemare et al., 2013) (where many algorithms like DQN (Mnih et al., 2013) model the environments as infinite-horizon discounted MDPs) such as Breakout, the agent cannot observe

states where most of the bricks are knocked out at the initial stage of the trajectory. Therefore, the agent still can only push forward the "information frontier" a few steps per deployment. That said, it is possible reduce the deployment complexity lower bound in stationary MDPs by adding more assumptions, such as the initial state distribution providing good coverage over the entire state space, or all the states are reachable in the first few time steps. However, because these assumptions do not always hold and may overly trivialize the exploration problem, we will not consider them in our algorithm design. Besides, although our algorithms in the next section are designed for non-stationary MDPs, they can be extended to stationary MDPs by sharing covariance matrices, and we believe the analyses can also be extended to match the lower bound in Corollary 3.3.

# 4 Towards Optimal Deployment Efficiency

In this section we provide algorithms with deployment-efficiency guarantees that nearly match the lower bounds established in Section 3. Although our lower bound results in Section 3 consider non-Markov policies, our algorithms in this section only use Markov policies (or a mixture of Markov policies, in the arbitrary policy setting), which are simpler to implement and compute and are already near-optimal in deployment efficiency.

**Inspiration from Lower Bounds: a Layer-by-Layer Exploration Strategy** The linear dependence on $H$ in the lower bounds implies a possibly deployment-efficient manner to explore, which we call a layer-by-layer strategy: conditioning on sufficient exploration in previous $h - 1$ time steps, we can use $\mathrm{poly}(d)$ deployments to sufficiently explore the $h$-th time step, then we only need $H \cdot \mathrm{poly}(d)$ deployments to explore the entire MDP. If we can reduce the deployment cost in each layer from $\mathrm{poly}(d)$ to $\Theta(d)$ or even $\Theta(1)$, then we can achieve the optimal deployment efficiency. Besides, as another motivation, in Appendix C.4, we will briefly discuss the additional benefits of the layer-by-layer strategy, which will be useful especially in "Safe DE-RL". In Sections 4.1 and 4.2, we will introduce algorithms based on this idea and provide theoretical guarantees.

## 4.1 Deployment-Efficient RL with Deterministic Policies

---

**Algorithm 1:** Layer-by-Layer Batch Exploration Strategy for Linear MDPs Given Reward Function

---

1   **Input**: Failure probability $\delta > 0$, and target accuracy $\varepsilon > 0$, $\beta \leftarrow c_\beta \cdot dH\sqrt{\log(dH\delta^{-1}\varepsilon^{-1})}$
    for some $c_\beta > 0$, total number of deployments $K$, batch size $N$,

2   $h_1 \leftarrow 1$       // $h_k$ denotes the layer to explore in iteration $k$, for all $k \in [K]$

3   **for** $k = 1, 2, ..., K$ **do**

4     $Q_{h_k+1}^k(\cdot, \cdot) \leftarrow 0$ and $V_{h_k+1}^k(\cdot) = 0$

5     **for** $h = h_k, h_k - 1, ..., 1$ **do**

6       $\Lambda_h^k \leftarrow I + \sum_{\tau=1}^{k-1}\sum_{n=1}^N \phi_h^{\tau n}(\phi_h^{\tau n})^\top$,      $u_h^k(\cdot, \cdot) \leftarrow \min\{\beta \cdot \sqrt{\phi(\cdot,\cdot)^\top(\Lambda_h^k)^{-1}\phi(\cdot,\cdot)}, H\}$

7       $w_h^k \leftarrow (\Lambda_h^k)^{-1}\sum_{\tau=1}^{k-1}\sum_{n=1}^N \phi_h^{\tau n} \cdot V_{h+1}^k(s_{h+1}^{\tau n})$

8       $Q_h^k(\cdot,\cdot) \leftarrow \min\{(w_h^k)^\top\phi(\cdot,\cdot) + r_h(\cdot,\cdot) + u_h^k(\cdot,\cdot), H\}$ and $V_h^k(\cdot) = \max_{a\in\mathcal{A}} Q_h^k(\cdot, a)$

9       $\pi_h^k(\cdot) \leftarrow \arg\max_{a\in\mathcal{A}} Q_h^k(\cdot, a)$

10    **end**

11    Define $\pi^k = \pi_1^k \circ \pi_2^k ... \circ \pi_{h_k}^k \circ \mathrm{unif}_{[h_k+1:H]}$

12    **for** $n = 1, ..., N$ **do**

13      Receive initial state $s_1^{kn} \sim d_1$

14      **for** $h = 1, 2, ..., H$ **do** Take action $a_h^{kn} \leftarrow \pi_h^k(s_h^{kn})$ and observe $s_{h+1}^{kn} \sim P_h(s_h^k, a_h^k)$ ;

15    **end**

16    Compute $\Delta_k \leftarrow \frac{2\beta}{N}\sum_{n=1}^N\sum_{h=1}^{h_k}\sqrt{\phi(s_h^{kn}, a_h^{kn})^\top(\Lambda_h^k)^{-1}\phi(s_h^{kn}, a_h^{kn})}$.

17    **if** $\Delta_k \geq \frac{\varepsilon h_k}{2H}$ **then** $h_{k+1} \leftarrow h_k$ ;

18    **else if** $h_k = H$ **then return** $\pi^k$ ;

19    **else** $h_{k+1} \leftarrow h_k + 1$ ;

20 **end**

---

In this sub-section, we focus on the setting where each deployed policy is deterministic. In Alg 1, we propose a provably deployment-efficient algorithm built on Least-Square Value Iteration with UCB (Jin et al., 2019)[4] and the "layer-by-layer" strategy. Briefly speaking, at deployment $k$, we focus on exploration in previous $h_k$ layers, and compute $\pi_1^k, \pi_2^k, ..., \pi_{h_k}^k$ by running LSVI-UCB in an MDP truncated at step $h_k$. After that, we deploy $\pi^k$ to collect $N$ trajectories, and complete the trajectory after time step $h_k$ with an arbitrary policy. (In the pseudocode we choose uniform, but the choice is inconsequential.) In line 19, we compute $\Delta_k$ with samples and use it to judge whether we should move on to the next layer till all $H$ layers have been explored. The theoretical guarantee is listed below, and the missing proofs are deferred to Appendix C.

**Theorem 4.1** (Deployment Complexity). *For arbitrary $\varepsilon, \delta > 0$, and arbitrary $c_K \geqslant 2$, as long as*

$N \geqslant c\Big(c_K \frac{H^{4c_K+1}d^{3c_K}}{\varepsilon^{2c_K}} \log^{2c_K}\big(\frac{Hd}{\delta\varepsilon}\big)\Big)^{\frac{1}{c_K-1}}$, *where $c$ is an absolute constant, by choosing*

$$K = c_K dH + 1. \tag{1}$$

*Algorithm 1 will terminate at iteration $k \leqslant K$ and return us a policy $\pi^k$, and with probability $1 - \delta$,*
$\mathbb{E}_{s_1 \sim d_1}[V_1^*(s_1) - V_1^{\pi^k}(s_1)] \leqslant \varepsilon$.

As an interesting observation, Eq (1) reflects the trade-off between the magnitude of $K$ and $N$ when $K$ is small. To see this, when we increase $c_K$ and keep it at the constant level, $K$ definitely increases while $N$ will be lower because its dependence on $d, H, \varepsilon, \delta$ decreases. Moreover, the benefit of increasing $c_K$ is only remarkable when $c_K$ is small (e.g. we have $N = O(H^9 d^6 \varepsilon^{-4})$ if $c_K = 2$, while $N = O(H^5 d^{3.6} \varepsilon^{-2.4})$ if $c_K = 6$), and even for moderately large $c_K$, the value of $N$ quickly approaches the limit $\lim_{c_K \to \infty} N = c\frac{H^4 d^3}{\varepsilon^2} \log^2(\frac{Hd}{\delta\varepsilon})$. It is still an open problem that whether the trade-off in Eq.1 is exact or not, and we leave it for the future work.

Another key step in proving the deployment efficiency of Alg. 1 is Lem. 4.2 below. In fact, by directly applying Lem. 4.2 to LSVI-UCB (Jin et al., 2019) with large batch sizes, we can achieve $O(dH)$ deployment complexity in deterministic policy setting without exploring in a layer-by-layer manner. We defer the discussion and the additional benefit of layer-by-layer strategy to Appx. C.4.

**Lemma 4.2.** *[Batched Finite Sample Elliptical Potential Lemma] Consider a sequence of matrices* $\mathbf{A}_0, \mathbf{A}_N, ..., \mathbf{A}_{(K-1)N} \in \mathbb{R}^{d \times d}$ *with* $\mathbf{A}_0 = I_{d \times d}$ *and* $\mathbf{A}_{kN} = \mathbf{A}_{(k-1)N} + \Phi_{k-1}$, *where* $\Phi_{k-1} = \sum_{t=(k-1)N+1}^{kN} \phi_t \phi_t^\top$ *and* $\max_{t \leqslant KN} \|\phi_t\| \leqslant 1$. *We define:* $\mathcal{K}^+ := \Big\{k \in [K] \Big| Tr(\mathbf{A}_{(k-1)N}^{-1} \Phi_{k-1}) \geqslant N\varepsilon\Big\}$. *For arbitrary $\varepsilon < 1$, and arbitrary $c_K \geqslant 2$, if $K = c_K dH + 1$, by choosing $N \geqslant$*
$c\Big(c_K \frac{Hd^{c_K}}{\varepsilon^{c_K}} \log^{c_K}\big(\frac{Hd}{\varepsilon}\big)\Big)^{\frac{1}{c_K-1}}$, *where $c$ is an absolute constant independent with $c_K, d, H, \varepsilon$, we have $|\mathcal{K}^+| \leqslant c_K d < K/H$.*

**Extension to Reward-free setting** Based on the similar methodology, we can design algorithms for reward-free setting (Wang et al., 2020b) and obtain $O(dH)$ deployment complexity. We defer the algorithms and proofs to Appx. D, and summarize the main result in Thm. D.4.

## 4.2 DEPLOYMENT-EFFICIENT RL WITH ARBITRARY POLICIES

From the discussion of lower bounds in Section 3, we know that in order to reduce the deployment complexity from $\Omega(dH)$ to $\widetilde{\Omega}(H)$, we have to utilize stochastic (and possibly non-Markov) policies and try to explore as many different directions as possible in each deployment (as opposed to 1 direction in Algorithm 1). The key challenge is to find a stochastic policy—before the deployment starts—which can sufficiently explore $d$ independent directions.

In Alg. 2, we overcome this difficulty by a new covariance matrix estimation method (Alg. 6 in Appx. E). The basic idea is that, for arbitrary policy $\pi$ [5], the covariance matrix $\Lambda_h^\pi := \mathbb{E}_\pi[\phi\phi^\top]$ can

---

[4]In order to align with the algorithm in reward-free setting, slightly different from (Jin et al., 2019) but similar to (Wang et al., 2020b), we run linear regression on $P_h V_h$ instead of $Q_h$.

[5]Here we mainly focus on evaluating deterministic policy or stochastic policy mixed from a finite number of deterministic policies, because for the other stochastic policies, exactly computing the expectation over policy distribution may be intractable.

---

**Algorithm 2:** Deployment-Efficient RL with Covariance Matrix Estimation

---

1 **Input**: Accuracy level $\varepsilon$; Iteration number $i_{\max}$; Resolution $\varepsilon_0$; Reward $r$; Bonus coefficient $\beta$.
2 **for** $h = 1, 2, ..., H$ **do**
3      Initialize $\pi_{h,1}$ with an arbitrary deterministic policy ; $\widetilde{\Sigma}_{h,1} = 2I, \Pi_h = \{\}$.
4      **for** $i = 1, 2, ..., i_{\max}$ **do**
5          $\widehat{\Lambda}_h^{\pi_{h,i}} \leftarrow \text{EstimateCovMatrix}(h, D_{[1:h-1]}, \Sigma_{[1:h-1]}, \pi_{h,i})$    # Alg 6, Appx E
6          $\widetilde{\Sigma}_{h,i+1} = \widetilde{\Sigma}_{h,i} + \widehat{\Lambda}_h^{\pi_{h,i}}$
7          $V_{h,i+1}, \bar{\pi}_{h,i+1} \leftarrow \text{SolveOptQ}(h, D_{[1:h-1]}, \Sigma_{[1:h-1]}, \beta, \widetilde{\Sigma}_{h,i+1}, \varepsilon_0)$ # Alg 5, Appx E
8          **if** $V_{h,i+1} \leqslant 3\nu_{\min}^2/8$ **then** break ;
9          $\Pi_h = \Pi_h \bigcup \{\bar{\pi}_{h,i+1}\}$
10      **end**
11      $\Sigma_h = I, D_h = \{\}, \pi_{h,\text{mix}} := \text{unif}(\Pi_h)$
12      **for** $n = 1, 2, ..., N$ **do**
13          Sample trajectories with $\pi_{h,\text{mix}}$
14          $\Sigma_h = \Sigma_h + \phi(s_{h,n}, a_{h,n})\phi(s_{h,n}, a_{h,n})^\top, \quad D_h = D_h \bigcup \{s_{h,n}, a_{h,n}, r_{h,n}, s_{h+1,n}\}$
15      **end**
16 **end**
17 **return** $\widehat{\pi}_r \leftarrow \text{Alg } 4(H, \{D_1, ..., D_H\}, r)$

---

be estimated element-wise by running policy evaluation for $\pi$ with $\phi_i \phi_j$ as a reward function, where $i, j \in [d]$ and $\phi_i$ denotes the $i$-th component of vector $\phi$.

However, a new challenge emerging is that, because the transition is stochastic, in order to guarantee low evaluation error for all possible policies $\bar{\pi}_{h,i+1}$, we need an union bound over all policies to be evaluated, which is challenging if the policy class is infinite. To overcome this issue, we discretize the value functions in Algorithm 5 (see Appendix E) to allow for a union bound over the policy space: after computing the Q-function by LSVI-UCB, before converting it to a greedy policy, we first project it to an $\varepsilon_0$-net of the entire Q-function class. In this way, the number of policy candidates is finite and the projection error can be controlled as long as $\varepsilon_0$ is small enough.

Using the above techniques, in Lines 3-10, we repeatedly use Alg 6 to estimate the accumulative covariance matrix $\widetilde{\Sigma}_{h,i+1}$ and further eliminate uncertainty by calling Alg 5 to find a policy (approximately) maximizing uncertainty-based reward function $\widetilde{R} := \|\phi\|_{\widetilde{\Sigma}_{h,i+1}^{-1}}$. For each $h \in [H]$, inductively conditioning on sufficient exploration in previous $h - 1$ layers, the errors of Alg 6 and Alg 5 will be small, and we will find a finite set of policies $\Pi_h$ to cover all dimensions in layer $h$. (This is similar to the notion of "policy cover" in Du et al. (2019); Agarwal et al. (2020a).) Then, layer $h$ can be explored sufficiently by deploying a uniform mixture of $\Pi$ and choosing $N$ large enough (Lines 11-15). Also note that the algorithm does not use the reward information, and is essentially a reward-free exploration algorithm. After exploring all $H$ layers, we obtain a dataset $\{D_1, ..., D_H\}$ and can use Alg 4 for planning with any given reward function $r$ satisfying Assump. A to obtain a near-optimal policy.

**Deployment complexity guarantees** We first introduce a quantity denoted as $\nu_{\min}$, which measures the reachability to each dimension in the linear MDP. In Appendix E.8, we will show that the $\nu_{\min}$ is no less than the "explorability" coefficient in Definition 2 of Zanette et al. (2020) and $\nu_{\min}^2$ is also lower bounded by the maximum of the smallest singular value of matrix $\mathbb{E}_\pi[\phi\phi^\top]$.

**Definition 4.3** (Reachability Coefficient)**.**

$$\nu_h := \min_{\|\theta\|=1} \max_\pi \sqrt{\mathbb{E}_\pi[(\phi_h^\top \theta)^2]} \ ; \qquad \nu_{\min} = \min_{h \in [H]} \nu_h \ .$$

Now, we are ready to state the main theorem of this section, and defer the formal version and its proofs to Appendix E. Our algorithm is effectively running reward-free exploration and therefore our results hold for arbitrary linear reward functions.

**Theorem 4.4.** *[Informal] For arbitrary $0 < \varepsilon, \delta < 1$, with proper choices of $i_{\max}, \varepsilon_0, \beta$, we can choose $N = \text{poly}(d, H, \frac{1}{\varepsilon}, \log\frac{1}{\delta}, \frac{1}{\nu_{\min}})$, such that, after $K = H$ deployments, with probability $1 - \delta$,*

*Algorithm 2 will collect a dataset $D = \{D_1, ..., D_H\}$, and if we run Alg 4 with $D$ and arbitrary reward function satisfying Assump. A, we will obtain $\widehat{\pi}_r$ such that $V_1^{\widehat{\pi}_r}(s_1; r) \geqslant V_1^*(s_1; r) - \varepsilon$.*

**Proof Sketch**   Next, we briefly discuss the key steps of the proof. Since $\varepsilon_0$ can be chosen to be very small, we will ignore the bias induced by $\varepsilon_0$ when providing intuitions. Our proof is based on the induction condition below. We first assume it holds after $h - 1$ deployments (which is true when $h = 1$), and then we try to prove at the $h$-th deployment we can explore layer $h$ well enough so that the condition holds for $h$.

**Condition 4.5.** [Induction Condition] Suppose after $h - 1$ deployments, we have the following induction condition for some $\xi < 1/d$, which will be determined later:

$$\max_\pi \mathbb{E}_\pi [\sum_{\widetilde{h}=1}^{h-1} \sqrt{\phi(s_{\widetilde{h}}, a_{\widetilde{h}})^\top \Sigma_{\widetilde{h}}^{-1} \phi(s_{\widetilde{h}}, a_{\widetilde{h}})}] \leqslant \frac{h-1}{H}\xi. \tag{2}$$

The l.h.s. of Eq.(2) measures the uncertainty in previous $h - 1$ layers after exploration. As a result, with high probability, the following estimations will be accurate:

$$\|\widehat{\Lambda}^{\pi_{h,i}} - \mathbb{E}_{\pi_{h,i}}[\phi(s_h, a_h)\phi(s_h, a_h)^\top]\|_{\infty,\infty} \leqslant O(\xi), \tag{3}$$

where $\| \cdot \|_{\infty,\infty}$ denotes the entry-wise maximum norm. This directly implies that:

$$\|\widetilde{\Sigma}_{h,i+1} - \Sigma_{h,i+1}\|_{\infty,\infty} \leqslant i \cdot O(\xi).$$

where $\Sigma_{h,i+1} := 2I + \sum_{i'=1}^{i} \mathbb{E}_{\pi_{h,i'}}[\phi(s_h, a_h)\phi(s_h, a_h)^\top]$ is the target value for $\widetilde{\Sigma}_{h,i+1}$ to approximate. Besides, recall that in Algorithm 5, we use $\sqrt{\phi^\top \widetilde{\Sigma}_{h,i+1}^{-1} \phi}$ as the reward function, and the induction condition also implies that:

$$|V_{h,i+1} - \max_\pi \mathbb{E}_\pi[\|\phi(s_h, a_h)\|_{\widetilde{\Sigma}_{h,i+1}^{-1}}]| \leqslant O(\xi).$$

As a result, if $\xi$ and the resolution $\varepsilon_0$ are small enough, $\bar{\pi}_{h,i+1}$ would gradually reduce the uncertainty and $V_{h,i+1}$ (also $\max_\pi \mathbb{E}_\pi[\|\phi(s_h, a_h)\|_{\widetilde{\Sigma}_{h,i+1}^{-1}}]$) will decrease. However, the bias is at the level $O(\xi)$, and therefore, no matter how small $\xi$ is, as long as $\xi > 0$, it is still possible that the policies in $\Pi_h$ do not cover all directions if some directions are very difficult to reach, and the error due to such a bias will be at the same level of the required accuracy in induction condition, i.e. $O(\xi)$. This is exactly where the "reachability coefficient" $\nu_{\min}$ definition helps. The introduction of $\nu_{\min}$ provides a threshold, and as long as $\xi$ is small enough so that the bias is lower than the threshold, each dimension will be reached with substantial probability when the breaking criterion in Line 9 is satisfied. As a result, by deploying $\mathrm{unif}(\Pi_h)$ and collecting a sufficiently large dataset, the induction condition will hold till layer $H$. Finally, combining the guarantee of Alg 4, we complete the proof.

## 5   CONCLUSION AND FUTURE WORK

In this paper, we propose a concrete theoretical formulation for DE-RL to fill the gap between existing RL literatures and real-world applications with deployment constraints. Based on our framework, we establish lower bounds for deployment complexity in linear MDPs, and provide novel algorithms and techniques to achieve optimal deployment efficiency. Besides, our formulation is flexible and can serve as building blocks for other practically relevant settings related to DE-RL. We conclude the paper with two such examples, defer a more detailed discussion to Appendix F, and leave the investigation to future work.

**Sample-Efficient DE-RL**   In our basic formulation in Definition 2.1, we focus on minimizing the deployment complexity $K$ and put very mild constraints on the per-deployment sample complexity $N$. In practice, however, the latter is also an important consideration, and we may face additional constraints on how large $N$ can be, as they can be upper bounded by e.g. the number of customers or patients our system is serving.

**Safe DE-RL**   In real-world applications, safety is also an important criterion. The definition for safety criterion in Safe DE-RL is still an open problem, but we believe it is an interesting setting since it implies a trade-off between exploration and exploitation in deployment-efficient setting.

## ACKNOWLEDGEMENTS

JH's research activities on this work were completed by December 2021 during his internship at MSRA. NJ acknowledges funding support from ARL Cooperative Agreement W911NF-17-2-0196, NSF IIS-2112471, and Adobe Data Science Research Award.

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
