# OpenReview forum: "Towards Deployment-Efficient Reinforcement Learning: Lower Bound and Optimality"
_ICLR.cc/2022/Conference — ICLR 2022 Spotlight_

### Official Review · Reviewer_Ckdx · 2021-10-29

**Correctness:** 4
**Technical Novelty And Significance:** 4
**Empirical Novelty And Significance:** Not applicable
**Recommendation:** 8
**Confidence:** 3

**Main Review:**

Strengths:

-- Deployment-efficiency is an impactful research area.

-- The derivations and algorithms are novel, and provide insights that may be useful to future theoretical or empirical works.

-- The writing is clear and easy to understand.

-- I did not look over the proofs carefully, but I found no issue with what I skimmed through.


Weaknesses/Questions:

-- I may have missed this, but it would help if you had some discussion comparing your sample & deployment complexity results to existing complexity results in the literature. For example, it is interesting to note that the sample complexity per deployment (N) in your Thm 4.1 is always at least as large as the sample complexity for finding an epsilon-opt policy (ignoring deployments) from Jin et al. 2019. These sorts of remarks help illuminate the sample complexity trade-offs introduced by the deployment-efficiency constraint.

-- In Section 4.1, I don't fully understand the sentence "To see this, when we increase..." Is there a typographical error here? It seems K and RHS of Eq 1 both refer to the same expression.

-- In Section 4.2, can you clarify what you mean by "uniform mixture of Pi"? It is not clear if you are uniformly mixing over Pi at each state, or sampling a policy from Pi at the beginning of each trajectory. If it is the latter, then I feel it is debatable whether this should count as a single deployment.

-- The work is wholly theoretical in nature. While I understand that there is limited space, some sort of empirical demonstration would be nice.

**Summary Of The Paper:**

The paper presents a theoretical perspective on deployment efficiency in linear MDPs. The paper formalizes the notion of deployment complexity and presents an information-theoretic lower bound for worst-case deployment complexity of any algorithm, identifying horizon as the main bottleneck for deployment efficiency (in addition to feature dimension when restricting to deterministic policies). The paper then presents two algorithms (for deterministic or stochastic policies) matching the lower bounds.

**Summary Of The Review:**

Overall, this paper provides a significant contribution to the field. I found the theoretical results insightful and, although the present submission contains no empirical results, I believe the derivations may inspire future more practical algorithms.

---

> ### Author Response · Authors · 2021-11-14
> **Response to Reviewer Ckdx**
>
> We thank the reviewer for the comments.
>
> -- Comparison of sample & deployment complexity results
>
> Thanks for this suggestion. We have included a discussion at the end of Appendix A (see also Table 1) in our revision.
>
> -- “To see this...” in Section 4.1
>
> Thanks for pointing out the typo, and we have fixed it in the revision.
>
> -- “Uniform mixture in Pi” in Section 4.2
>
> It means sampling a policy from Pi uniformly at the beginning of the episode. We understand your concern and have actually given some discussion to this issue in the paper (see the top of page 2 and the top of page 4). In particular, the fundamental barrier in deployment-efficient RL is the limited adaptivity, that our algorithm cannot update the policy on the fly using the latest information collected online in a fully timely manner. (Previous works in related topics, e.g., low switching cost, have made similar points.) Our uniform policy still respects this restriction, as the policy is fully specified before each deployment and is not updated within the deployment. Also, a mixture policy can be simply viewed as a single non-Markov stochastic policy, as mentioned in the first paragraph on page 4.
>
> From a practical perspective, there are many application scenarios where such a form of mixture policies is indeed implementable. This is especially the case when our system serves a population of users: as mentioned in the first paragraph of page 2, “randomized experiments on a population of users can be viewed as deploying a mixture of deterministic policies”. That said, we also admit that sometimes, implementing such mixture policies can be more difficult than implementing a usual (Markov) policy or infeasible, which is part of the reason why we study the two settings (deploying deterministic vs mixture policies) separately and “believe both are … of interest” (page 2).

---

### Official Review · Reviewer_iHZz · 2021-11-01

**Correctness:** 4
**Technical Novelty And Significance:** 3
**Empirical Novelty And Significance:** Not applicable
**Recommendation:** 8
**Confidence:** 4

**Main Review:**

This paper studies deployment efficient RL from the theoretical perspective. It shows when MDP is linear, the lower bound of deployment complexity for deterministic policy and arbitrary policy is $\tilde{\Omega}(dH)$ and $\Omega(H)$ respectively, and Layer-by-Layer Batch Exploration Strategy for linear MDPs and Deployment-Efficient RL with Covariance Matrix Estimation can achieve this deployment complexity. Those results are new and answer the questions that are left by [Gao et al.] (where global switching cost is considered). I believe this result is of great importance to the RL community.

Weakness: Theorem 4.1 has exponent $c_K$ in it, where $c_K$ can be any arbitrary constant. Could you choose a specific constant $c_K$ so that the bound gives you the best trade-off between $N$ and $K$? say the total sample complexity $N\cdot K$ to be small?




**Summary Of The Paper:**

This paper studies deployment-efficient reinforcement learning and provides a theoretical perspective. In this setting, this paper provides lower bounds and upper bounds respectively for determinisitc policies and arbitrary policies when the MDPs have linear structures. The deployment complexity is near-optimal.

**Summary Of The Review:**

This paper provides a solid study in deployment efficient RL from the theoretical perspective. Therefore, I choose acceptance.

---

> ### Author Response · Authors · 2021-11-14
> **Response to Reviewer iHZz**
>
> We thank the reviewer for the comments.
>
> As for Theorem 4.1, as we increase $c_K$, the dependence of $N$ on $d$, $H$ and $\epsilon^{-1}$ will decrease and the asymptotic (i.e., the best possible) dependence would be $O(H^{4}d^{3}\epsilon^{-2})$. But we cannot increase $c_K$ to infinity since it will result in a large $K$ and therefore, a large sample complexity, as implied by our Theorem 4.1. Fortunately, a mildly large $c_K$ can already approximately achieve the asymptotic dependence $N$. For example, if we choose $c_K = 100$, then $N \approx O(H^{4.05}d^{3.03}\epsilon^{-2.02})$, which is almost the same order of the asymptotic result, while $K=100dH+1$ can still be regarded as $O(dH)$. To summarize, choosing a mildly large constant for $c_K$ already gives a near-optimal sample complexity, and any other choice of $c_K$ cannot make any significant improvement to the bound.

---

> > ### Comment · Reviewer_iHZz · 2021-11-22
> > **Response to the authors**
> >
> > I think the authors for the detailed reply. The current result is already good enough and but if it can be improved to have $\epsilon^2$ for any $c_K$ then I think the result would be much stronger, regardless of the order in other parameters (since $\epsilon^2$ is the standard statistical rate). Could you explain in which part of your proof makes the $\epsilon$ in the bound also depend on $c_K$?
> >
> > Anyway, I will keep my original score as acceptance.

---

> > > ### Author Response · Authors · 2021-11-23
> > > **Thanks for this interesting question**
> > >
> > > Indeed it would be nicer to have $N=O(1/\epsilon^2)$ even with a small constant $c_K$. The technical difficulty that prevents us from having this result is in the Ellipsoid Potential Lemma 4.2; see its proof on page 26. In particular, in Equation above Eq.(7), we have $(...)^p \ge 1 + KN/d$, where $p$ is later set to $c_K$. In fact, it’s the linear dependence on $N$ in the RHS ($1+KN/d$) that gives rise to the complication that the order of $N$’s dependence on $\epsilon$ involves $c_K$; if the RHS did not contain N, we would have $N=O(1/\epsilon^2)$ in our final result (here we have a slight abuse of notation, and $\epsilon$ in this geometric lemma corresponds to $\epsilon^2$ in the main theorem). The $N$ in $1+KN/d$ comes from upper-bounding the determinant of $A_{KN}$ by its trace, which grows linearly with the number of items in the summation that defines $A_{KN}$.

---

### Official Review · Reviewer_5FHN · 2021-11-01

**Correctness:** 4
**Technical Novelty And Significance:** 3
**Empirical Novelty And Significance:** Not applicable
**Recommendation:** 8
**Confidence:** 4

**Main Review:**

Strengths:

- DE-RL is an interesting framework, reasonably motivated by real-world challenges.

- The lower bounds are reasonable and proofs are clear. The scenarios considered by the lower bounds are sufficiently general and the paper does a great job illustrating the intuition behind the bounds. The authors also commented on converting the nonstationary lower bounds into stationary lower bounds, showing that the deployment cost is somewhat inherent to MDPs rather inherent to nonstationary tasks.

- The motivations behind the algorithms' construction are clearly stated and the authors have shown the connection between this paper and existing literature.

- Assumptions for proving the upperbounds are reasonable and does not appear to be stronger than existing literature.

- Additional results in the appendix are comprehensive and clearly stated.

Weaknesses:

- Nitpick: Definition 2.1 could be improved by requiring $K$ to be on the same order of magnitude as the lower bound. In its current form, it would appear that a smaller $K$, in terms of constant terms, is more deployment efficient than existing algorithms. Allowing for $K$'s that match the lower bound up to constant factors would eliminate such potential ambiguity.

**Summary Of The Paper:**

The paper focuses on the theoretical properties of Deployment Efficient Reinforcement Learning, DE-RL, focusing on linear MDPs. Deployment complexity is defined as the number of times that a different policy is selected to collect data, and an algorithm is said to be deployment efficient if the number of deployments is as small as possible and the number of trajectories per iteration is polynomial.

In section 3, the authors provide a lower bound for the task, under both deterministic and stochastic policies.

In section 4, the authors provide detailed algorithms for DE-RL, as well as relevant upper bounds. In the deterministic case the upperbound matches lower bound up to constant factors, and in the stochastic case the upperbound matches the lower bound *up to log factors* and as long as $\nu_{\textrm{min}}$ is bounded.

**Summary Of The Review:**

The paper introduces a new framework, DE-RL, and provides lower bounds under different settings. By drawing inspiration from reward-free and provably efficient RL, the paper provides efficient algorithms for the task.

The setting is interesting and motivated by real-world concerns.The results are comprehensive and reasonable, with clearly explained proof in the appendix.

---

> ### Author Response · Authors · 2021-11-14
> **Response to Reviewer 5FHN**
>
> We thank the reviewer for the comments.
>
> We agree. In fact, after reflecting on your comment, we realize that it may be better to mathematically define what deployment complexity is (rather than what deployment-efficient RL is), where the issue of minimizing $K$ does not arise. Deployment-efficient RL is simply about designing algorithms with provable guarantees of low deployment complexity. This is analogous to the situation in sample-efficient RL: we have formal definitions of what sample complexity is; sample-efficient RL is simply about designing algorithms with low sample complexity and is more of an informal term.

---

### Official Review · Reviewer_sYkB · 2021-11-05

**Correctness:** 4
**Technical Novelty And Significance:** 3
**Empirical Novelty And Significance:** Not applicable
**Recommendation:** 8
**Confidence:** 3

**Main Review:**

Strengths:
+ The paper is well motivated and timely, it considers the problem of minimizing cycles of interaction of an agent with the environment which still learning near-optimal policies.
+ The authors prove a novel lower bound on the deployment complexity in the linear MDP setting and show that this is achievable.
+ A framework of optimization with constraints is proposed to prove the above results; the applications of safe and sample-efficient RL are highlighted in this context.

Weaknesses:
+ No practical realizations of the theoretical results are demonstrated.

**Summary Of The Paper:**

The authors consider the problem of deployment complexity in reinforcement learning problems, where we want to reduce the number of cycles an agent is typically deployed for interacting with the environment. In the context of a linear MDP, the authors prove an information theoretic lower bound. Further, they also propose algorithms which achieve this optimal deployment efficiency.

**Summary Of The Review:**

The paper makes novel theoretical contributions to the challenge of deployment costs in practical RL algorithms. The problem is well motivated, well written and good fit for the venue.

---

> ### Author Response · Authors · 2021-11-14
> **Response to Reviewer sYkB**
>
> We thank the reviewer for the comments. We agree that empirical evaluation would be valuable and plan to investigate in subsequent works, and focus on laying the theoretical foundations of deployment-efficient RL in this work.

---

### Author Response · Authors · 2021-11-14
**General Comments on Revisions**

We have revised the paper and marked the changes in red. Apart from small typo fixes and minor representation changes, the main change is the addition of Table 1 in Appendix A (and the associated text), where we compare the sample and deployment complexities of our algorithms to those of existing methods which did not consider the deployment constraint in our Def. 2.1. The goal is to investigate the trade-off between deployment complexity and sample complexity, as suggested by Reviewers iHZz and Ckdx.

---

### Decision · Program_Chairs · 2022-01-20

**Decision:**

Accept (Spotlight)

**Comment:**

The authors’ present a precise definition of deployment efficient RL, where each new update of the policy may be costly, and theoretically analyze this for finite-horizon linear MDPs. The authors include an information-theoretic lower bound for the number of deployments required. The reviewers found this an important setting of interest and appreciated the theoretical contributions. The authors’ carefully addressed the raised points and also addressed questions about deployment complexity and sample complexity in their revised work. One weakness of the paper is that it does not provide empirical results and the linear MDP assumption, while quite popular in theoretical RL over the last few years, is quite restrictive. However,the paper still provides a very interesting theoretical contribution for an important topic and I recommend acceptance.